



# Lake area and volume variation in the endorheic basin of
# the Tibetan Plateau from 1989 to 2019
Liuming Wang[1,3], Junxiao Wang[2,3], Mengyao Li[1], Liping Zhu[4], Xingong Li[5]
[1]Schoole of Geography and Ocean Science, Nanjing University, Nanjing, 210023, China
[2]Schoole of Public Administration, University of Finance & Economics, Nanjing, 210023, China
[3]Laboratory of Coastal Zone Exploitation and Protection, Ministry of Natural Resource, Nanjing, Jiangsu,
210017, China
[4]Key Laboratory of Tibetan Environment Changes and Land Surface Processes (TEL), Institute of
Tibetan Plateau Research (ITP), Chinese Academy of Sciences, Beijing 100101, China
[5]Department of Geography & Atmospheric Science, University of Kansas, Lawrence, 66045, United
States of America
*Correspondence to*: Junxiao Wang(wangjunxiao@nufe.edu.cn) and Xingong Li (lixi@ku.edu)
**Abstract.** The Tibetan Plateau, known as "the third pole of the Earth", is a region susceptible to
climate change. With little human disturbance, lake storage changes serve as a unique indicator of
climate change, but comprehensive lake storage data are rare in the region, especially for the lakes with
an area less than 10 km$^2$ which are the most sensitive to environmental changes. In this paper, we
completed a census of annual lake volume change for 976 lakes larger than 1 km$^2$ in the endorheic
basin of the Tibetan Plateau (EBTP) during 1989-2019 using Landsat imagery and digital terrain
models. Validation and comparison with several existing studies indicate that our data are more
reliable. Lake volume in the EBTP exhibited a net increase of 193.45 km$^3$ during the time period with
an increasing rate of 6.45 km$^3$ year$^{-1}$. In general, the larger the lake area, the greater the lake volume
change, though there are some exceptions. Lakes with an area less than 10 km$^2$ have more severe
volume change whether decreasing or increasing. This research complements existing lake studies by
providing a comprehensive and long-term lake volume change data for the region. The dataset is
available on Zenodo (https://doi.org/10.5281/zenodo.5543615, Wang et al., 2021).
**Keywords.** Tibetan Plateau, Landsat, relative lake volume
**1 Introduction**
Alpine lakes are susceptible to climate change in arid and semi-arid endorheic watersheds (Williamson
et al., 2009; Yao et al., 2018). One of the world's largest alpine lake groups are found in the Tibetan
Plateau (TP) (Yang et al., 2017a), which, together with its surrounding regions, is often referred to as
"the Third Pole of the Earth" (Qiu, 2008) and the "roof and the world" and provides vital water resources
for more than a billion population in Asia and is a sensitive region undergoing rapid climate change
(Field, 2014).
With little human disturbance in the region, lake volume variation may serve as an important indicator
that reflects regional hydrologic system's responses to climate change (Boos and Kuang, 2010; Yang et
al., 2017b). In the past 50 years, the TP has undergone a much faster warming trend (~0.447 °C per
decade) than the global average (0.15–0.20 °C per decade) (Hansen et al., 2010; Xu et al., 2008), which
posed inevitable impacts on the water budget of its alpine lakes (Lei et al., 2017; Liu et al., 2009). Lake
area in the TP has been increasing, which is the opposite of the changes in other regions of China (Ma et
al., 2010), Asia's plateaus (Zhang et al., 2017a), and other regions or drainage basins across the globe
(Donchyts et al., 2016). Furthermore, alpine lakes in the endorheic basin have a unique role as they serve
as nodes linking atmospheric, cryospheric, and biospheric components of the hydrological cycle. To
understand climate change forcing on regional hydrological cycles in the region, it is essential to monitor
the volume change of these alpine lakes (Song et al., 2014).
Due to the harsh environment and few in situ observations, satellite remote sensing has become an
indispensable tool for studying the dynamics of alpine lakes in the TP (Song et al., 2016; Song et al.,
2017; Wan et al., 2016). The advent of satellite imagery makes it possible for long-term and large-scale
monitoring of alpine lakes (Lei et al., 2017; Li et al., 2019; Song et al., 2016; Yang et al., 2017a; Yao et
al., 2018; Zhang et al., 2017b; Zhou et al., 2015) and lake volume changes in the TP have been examined
using Landsat data (Ma et al., 2010; Song et al., 2014; Zhang et al., 2017a). Table 1 summarizes recent
studies on lake volume changes in the region. In the two most recent studies, Li et al. (2019) examined
multiyear changes in water level and storage of 52 lakes with an area larger than 150 $km^2$ in the TP using
altimetry and optical remote sensing images during 2000–2017. Yao et al. (2018) integrated optical
imagery and digital elevation models and studied the lake water storage (LWS) change of 871 lakes from
2002–2015 in the Changtang Plateau (CP) of north-western TP. However, existing studies are limited to
either some large lakes, specific years (every 5 or 10 years), or only for a time span of less than 15 years.
According to existing research, there are about 1200 lakes with an area larger than 1 $km^2$ in the TP (Zhang
et al., 2017a; Zhang et al., 2020) and earth observation satellites, such as Landsat mission, span more
than 30 years (Huang et al., 2017). However, existing studies have neither made full use of existing earth



observation data, nor have they covered more than 75% of the lakes in the TP. Without a long-term
comprehensive census on lake volume change, it is difficult to study the impacts of climate change on
the hydrological system in the region.
**Table 1: Recent lake studies and datasets in the TP.**

| Study | No. of lakes | Temporal resolution | Timespan |
|---|---|---|---|
| Zhang et al. (2017b) | 60-70 | One record in the 1970s and annual for 1989–2015 | 1972-2015 |
| Yang et al. (2017b) | 114 | 1976, 1990, 2000, 2005 and 2013 | 1976-2013 |
| Yang et al. (2017a) | 874 | Monthly | 2009-2014 |
| Yao et al. (2018) | 871 | Annual | 2002-2015 |
| Li et al. (2019) | 52 | Monthly | 2000-2017 |
| **This study** | 976 | Annual | 1989-2019 |


In this research, using the Google Earth Engine (GEE) geospatial analysis platform, we analyzed Landsat
imagery in the past 30 years (1989 – 2019) to obtain annual lake area time series data for 976 lakes with
a maximum area larger than 1 km$^2$ in the endorheic basin of the TP (EBTP). We further derived the
relationship between lake area and surface elevation using digital terrain model data and estimated the
annual volume change for the lakes. This study provides so far the most comprehensive census on lake
volume change in the EBTP.
**2 Study area and data**
The endorheic basin of the TP (78.646E-99.379E, 29.829N-39.419N), which has a total area of 1.42 x
10$^6$ km$^2$, can be generally divided into two sub-basins: Inner and Qaidam basins (IB and QB) (Fig. 1).
Most lakes in the endorheic basin were expanding under climate change (Zhou et al., 2015). 976 lakes
with a maximum area larger than 1 km$^2$ were identified in this study, which had a total area of 30912.03
km$^2$ in 2019.

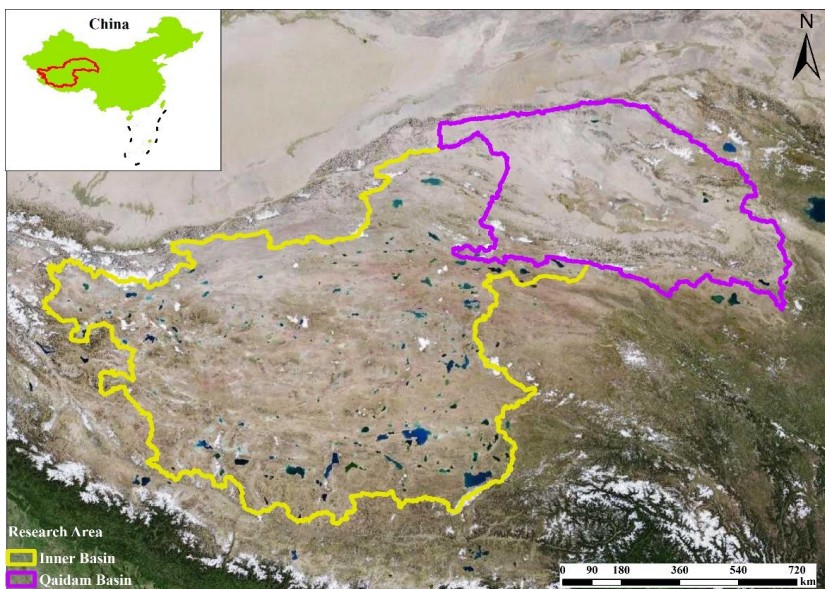

**Figure 1: Study area and two sub-regions (inner basin and Qaidam basin). Background remote sensing image is from http://t0.tianditu.gov.cn/img_c/wmts.**

The data used in this research include Landsat imagery, Joint Research Centre Global Surface Water (JRC-GSW) data, Shuttle Radar Topography Mission (SRTM) digital elevation mode (DEM)l, Advanced Land Observing Satellite (ALOS) digital surface model (DSM), and several public lake storage data. Imagery from Landsat-5 TM (1984-2012), Landsat-7 ETM+ (1999-), and Landsat-8 OLI (2013-) was used to extract lake and calculate annual lake area. The JRC-GSW data were generated using over 3 million scenes from Landsat 5, 7, and 8 acquired between 16 March 1984 and 31 December 2019 (Pekel et al., 2016). The dataset provides monthly surface water from 1984 to 2019 and statistics on the extent and change of surface water. The dataset was used to identify individual lakes and their analysis extents in this study. SRTM DEM and ALOS DSM (digital terrain model, DTM hereafter) were used to delineate lake's approximate extents from JRC-GSW data (see Sect. 3.1) and to establish the relationship between lake area and water surface elevation (see Sect. 3.4).

For validation purpose, we compared our results with a widely used lake surface elevation/storage data from the Laboratoire d'Etudes en Géophysique et Océanographie Spatiales (LEGOS) Hydroweb (Crétaux et al., 2011) and two most recent lake volume data from Li et al. (2019) and Yao et al. (2018). For these datasets, we used the overlapping lakes in the comparison.

## 3 Methods

In this research, calculating the lakes relative volume can be divided into two steps. The first step is to identify individual lakes, determine their analysis extents, and calculate annual lake area from Landsat imagery. The second step is to derive lake area-elevation relationship, estimate lake surface elevation from lake area, and calculate lake volume change. Details in the first step are shown in Fig. 2, which include three sub-steps: lake identification, analysis extent and seed determination (Sect. 3.1), water classification and segmentation (Sect. 3.2), and annual lake area calculation (Sect. 3.3). For the second step, Sect. 3.4 explains the way we construct the lake surface elevation-area relationship and Sect. 3.5 explains how to get the lake annual relative volume.

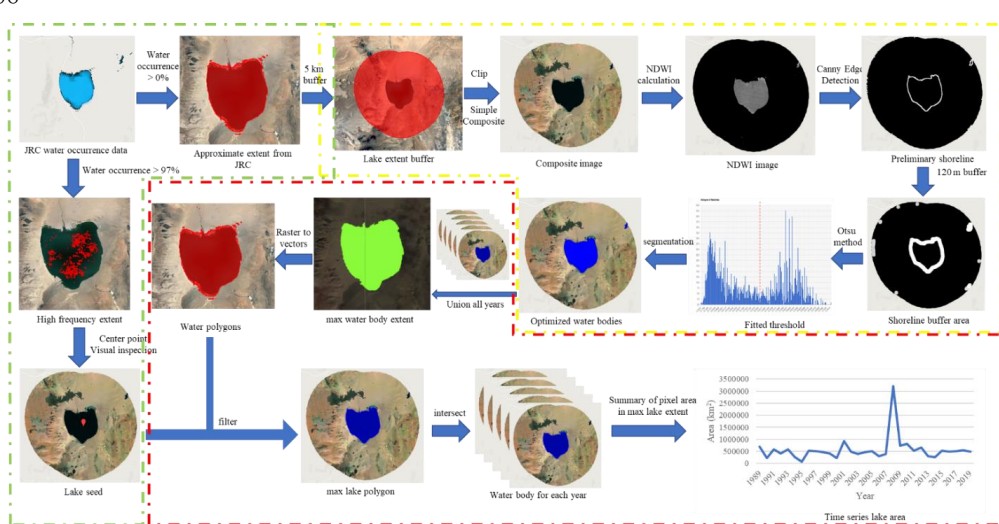

**Figure 2: Workflow for calculating annual lake area from Landsat imagery. Background remote sensing image is from http://t0.tianditu.gov.cn/img_c/wmts.**

### 3.1 Lake identification and analysis extent and seed determination

Due to the vast size of the EBTP and long term of Landsat imagery, we need to limit image processing to the lakes and their surrounding areas, so as to reduce computing resources and improve efficiency. For this purpose, we first need to identify the lakes and determine their analysis extents. Methods introduced in the following sections are all performed inside a lake's analysis extents.





We used the JRC-GSW data to identify the lakes in the study area. All the pixels with a positive water
cover frequency on the water occurrence band of the JRC-GSW data were retained, representing the
maximum water extent between 1984 and 2019. From those water pixels, spatially connected pixels were
identified as individual waterbodies and those with an area larger than 1 km$^2$ were kept. Some of those
waterbodies include both lakes and the rivers connected with them, especially for large lakes (Fig. 3).
The border between lakes and rivers is hard to define but we assume that the primary waterbody of a lake
is relatively flat and should have a slope close to zero. We used SRTM DEM to calculate the slope for
each waterbody pixel. Pixels with a slope greater than 0° are considered rivers and removed from the
waterbody. In this step, several patches of waterbody pixels may occur. We visually inspected those
patches and only kept the patch that represents the approximate extent of the lake associated with the
waterbody. This approach worked effectively for water bodies larger than 50 km$^2$ and the approximate
lake extents of 490 lakes were identified this way. In the process, we found there is a river linking two
lakes from high resolution remote sensing images (see Sect. 5.1 and Fig. 15). For these two lakes, the
linking river was kept and these two lakes were treat as one lake in our research. This situation happened
only once and these two lakes were usually treated as separate lakes in former reseach (Li et al., 2019;
Yao et al., 2018). The above procedure, however, tends to remove many small waterbodies entirely. So
for waterbodies less than 50 km$^2$, we inspected each waterbody visually and manually drew the
approximate lake extents, and we identified 486 more lakes and their approximate extents. Altogether,
we identified a total of 976 lakes and their approximate extents in the study area. Buffers of the
approximate lake extents were generated and used as analysis extents for the lakes so the accuracy of the
approximate lake extents is not an issue.

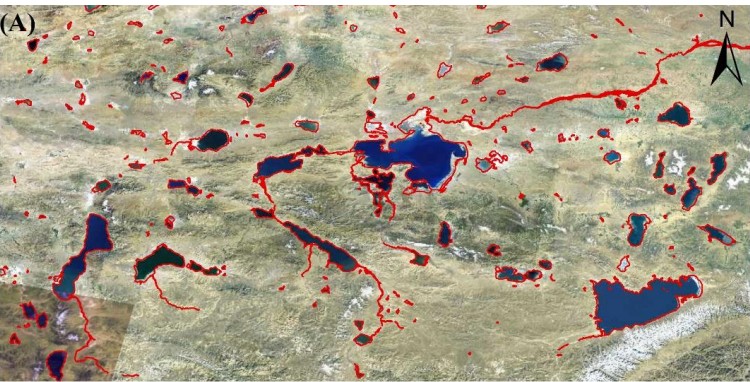

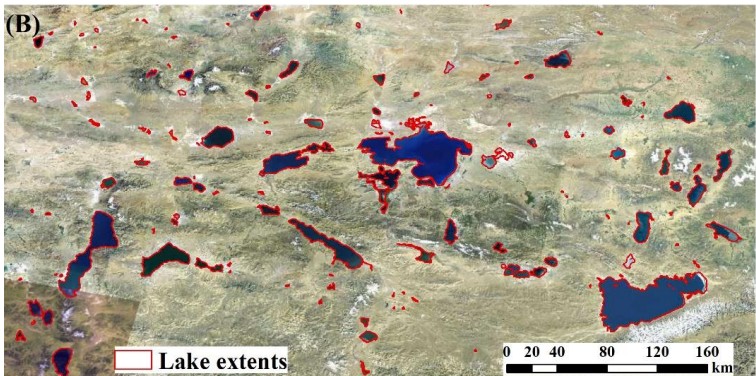

**Figure 3: Lakes identification and approximate extent determination. (A) Waterbodies with rivers; (B) Lake approximate extents after removing connected rivers. Background remote sensing image is from http://t0.tianditu.gov.cn/img_c/wmts.**

In addition to lake approximate extents, a point is created for each lake (hereafter lake seed) to identify and distinguish the target lake from other waterbodies within its analysis extent. The centroid point of each lake's approximate extent was calculated as the initial lake seed location but these points were manually checked and edited if necessary to make sure they are inside their lake approximate extents.

**3.2 Water segmentation**

Although the JRC-GSW data provide global monthly surface water map, it is not designed for mapping alpine lakes specifically. As such, we developed our own method for mapping lakes in the EBTP from Landsat imagery.

Based on the lake approximate extents obtained in Sect. 3.1, a 5 km buffer was generated around each extent and all the analyses hereafter in this section are confined to this analysis extent. Since there are more than 30,000 Landsat images in our study area within the study period, Google Earth Engine (GEE)





(Gorelick et al., 2017) was used for image processing and data analysis. We first selected Landsat images
between June and November in each year to exclude images with snow and ice. Landsat quality
assessment band (hereafter BQA band) was used to remove cloud, shadow, saturation (for Landsat 5, 7
and 8) and terrain occlusion (for Landsat 8 only) pixels on each image. A composite image was then
generated with the selected images using the SimpleComposite function in GEE. The function computes
a Landsat top of atmosphere (TOA) composite from a collection of raw Landsat scenes. It calculates a
cloud score (between 0 and 100) at each pixel for each image, selects the pixels with a cloud score less
than a certain threshold, and calculate a percentile pixel value for the composite image. In this research,
we used a cloud score threshold of 10 and a percentile value of 0. By using this function with the
parameters, we removed most cloud and generated annual max-water composite images. More details on
the function can be found at https://developers.google.com/earth-engine/guides/landsat#simple-
composite.
With the annual composite images, lake water pixels are classified using normalized difference water
index (NDWI) (Gao, 1996):
$$\text{NDWI} = \frac{B_G - B_{NIR}}{B_G + B_{NIR}} \tag{1}$$
where $B_G$, $B_{NIR}$ refer to green and near infrared bands, which is band 2 and 4 for Landsat 5/7 TM/ETM+
images and bands 3 and 5 for Landsat 8 OLI images, respectively. Several other indexes have been used
for lake mapping, such as modified NDWI (MNDWI) (Weekley and Li, 2019), normalized difference
moisture index (NDMI) (Elsahabi et al., 2016), and water ratio index (WRI) (Barbieux et al., 2018;
Elsahabi et al., 2016). We chose NDWI in this study as existing research indicated that NDWI appears
to be more robust in detecting lake extent under various water conditions (Qiao et al., 2019; Rokni et al.,

172 2014).

Thresholding (or segmentation) is a key step in extracting water pixels from NDWI images. Usually,
pixels with a NDWI value greater than 0 are considered as water. However, because of disparate
geographical environment and dynamic water conditions, it is impossible to use the same NDWI
threshold for all the lakes in all the years. In this research, we used local Otsu method (Otsu, 1979;
Setiawan et al., 2017) to dynamically segment NDWI images. Specifically, a Canny edge detection
algorithm (Bao et al., 2005) was first used to extract lake shorelines from NDWI images (see the yellow





box in Fig. 2). A 120-m double-sided buffer was then generated around the shorelines and Otsu method
was applied to obtain an optimal threshold that separates water from background pixels within the buffer.
This locally derived and image specific threshold was then used to extract lake pixels.
**3.3 Annual lake area**
As water level changes, some lakes may have several separate waterbodies in some years due to reduced
water volume. To handle this situation, we merged all the annual water pixels within a lake's analysis
extent and, from which, we then identified the spatially connected water pixels which contains the lake's
seed as the lake's maximum water extent during the study period. The maximum lake water extent is
then used to identify annual lake water pixels and calculate annual lake area (see red box in Fig. 2). In
this way, even if a lake has separate waterbodies in some of the years, all the waterbodies are counted as
parts of the same lake.
The Landsat imagery has several series, including Landsat-5 TM (1984-2012), Landsat-7 ETM+ (1999-),
and Landsat-8(Cristóbal et al., 2009). When imagery from multiple sensors (Landsat 5 & 7 and 7 & 8)
are available, lake area was calculated separately from each sensor and then combined. If the relative
difference between the sensors is within 2%, the average area is used for the year. Otherwise, annual
Landsat composite images and lake boundaries were manually examined to decide which area is more
accurate. In addition, annual lake area was manually checked if there is a significant change from
previous and following years. If the annual composite image is contaminated and unreliable, lake area
for the year was linearly interpolated using prior and later year's lake area. Through those steps, we
obtained the annual maximum lake area for each lake from Landsat imagery.
**3.4 Lake surface elevation**
Lake surface elevation is essential to calculate water volume change. Both satellite altimetry and DTM
have been used to estimate lake surface elevation (Li et al., 2019; Qiao et al., 2019; Song et al., 2014).
While satellite altimetry is more accurate, it is limited to less than 170 large lakes in the TP (Hwang et
al., 2019; Jiang et al., 2017; Li et al., 2017) and even fewer in our study area (Zhang et al., 2017b).
Because of this, we used DTM data to estimate lake surface elevation.

205 Without lake bathymetry data, we can only estimate lake surface elevation based on the elevation-area

206 relationship derived from DTM collected after 2000 assuming that the slope below lake surface is similar

207 to that above lake surface in 2000 (Yang et al., 2017b). Some commonly used methods include linear

208 equation (Yang et al., 2017b), second order parabolic equation (Li et al., 2019) and monotonic cubic

209 spline fitting (Yao et al., 2018). These methods have their own advantages and disadvantages. While the

210 linear interpolation is the simplest, more complicated methods such as the cubic spline interpolation,

211 which constructs polynomial functions, can fit data more smoothly (Gray et al., 2018). Linear regression

212 is usually suitable for elevation-area relationship with a fixed slope. And second order parabolic equation

213 is suitable for simulating the relationship with small changes in slope. The monotonic cubic spline fitting

214 can model the elevation-area relationship with large slope changes (Gray et al., 2018).

215 Although existing research indicates that monotonic cubic interpolation (MCI) has the best performance

216 in fitting elevation-area relationship (Yao et al., 2018), we found that MCI may overfit (see Sect. 5.2). In

217 this research, a combination of linear regression (LR), second order polynomial regression (SOPR), and

218 MCI methods was used to derive the elevation-area relationship which was then used to estimate surface

219 elevation based on lake area. The elevation-area pairs, where the elevation starts at from the lowest

220 elevation, stops at the highest elevation and increases at an interval of 1 m within each lake analysis

221 extent, were obtained from SRTM and ALOS separately. At each elevation, pixels with an elevation less

222 than the current elevation are kept and connected components are identified. The maximum lake water

223 extent (see 3.4) is then used to select the components belonging to the lake. The sum of all the components'

224 area is calculated as the area for the current elevation. The minimum (MinA) and maximum (MaxA)

225 annual lake area from Landsat are then used to select the elevation-area pairs whose area is in the range

226 of [MinA/1.5, MaxA*1.5] from both SRTM and ALOS, and the list with more elevation-area pairs is

227 kept. If the two lists have the same length, the SRTM list is kept. The choice of the data fitting methods

228 depends on the number of elevation-area pairs in the range of [MinA/1.5, MaxA*1.5], which is discussed

229 below and summarized in Table 2:

230 (1) If the number of data pairs is zero or one, we generated a new list of elevation-area pairs from the

231 selected DTM with eight pairs whose area starts with MaxA*1.5. The LR method was then used to derive

232 the elevation-area relationship (labelled LRN);





(2) If the number of data pairs is two, we directly used LR to derive the elevation-area relationship
(labelled LRC);
(3) If the number of data pairs is equal to or greater than five and lake area range from the selected DTM
fully covers the area range ([MinA, MaxA]) from Landsat imagery, the MCI method was used;
(4) In other cases, the SOPR method was used. If the symmetry axis of the SOPR model falls into [MinA,
MaxA], the elevation-area relationship will be non-monotonic (see Sect. 5.2). To avoid this, the
symmetry axis was calculated, and if the symmetry axis fell into [MinA, MaxA], LR method was used
instead (labeled LRS).
**Table 2: Selection of data fitting methods for deriving elevation- area relationship for each lake.**

| Conditions | Method | Abbreviation |
|---|---|---|
| The number of data pairs is 0 or 1 | Generate 8 new data pairs and then use LR | LRN |
| Number of data pairs = two | LR | LRC |
| Number of data pairs >= five and MinA is larger than the minimum area from DTM | MCI | MCI |
| None of the above | SOPR but use LR when the symmetry axis of SOPR falls into [MinA, MaxA] | SOPR / LRS |

**3.5 Lake volume**
While it is impossible to obtain lake water volume without bathymetry data (Crétaux et al., 2016), we
can calculate relative lake volume (RLV) between two dates with the lake area and elevation at those
dates. RLV from time t1 to time t2 can be calculated by the integral of an elevation-area relationship
function:
$$\text{RLV}_{t1-t2} = \int_{E_{t1}}^{E_{t2}} A dE = \int_{E_{t1}}^{E_{t2}} f(E) dE \qquad (2)$$
$$f(E) = A = a + bE + cE^2 \ or \ d + eE \qquad (3)$$
where E denotes lake surface elevation, and A is the lake area at the elevation. f(E) is the fitted elevation-
area function using the LR or SOPR methods, and a, b, c, d, and e are the coefficients of the SOPR and
LR models.
Since the MCI function is not integrable analytically, we cut the lake volume between two dates into
frustums with 1 m intervals in elevation (Fig. 4). With an elevation list $[E_{t1}, E_{t1} + 1, E_{t1} + 2, \dots, E_{t2} -$



$1, E_{t2}$], the corresponding lake area was obtained using a fitted MCI. The RLV is the sum of all the
frustums (i.e., $\sum_1^n V_{Fn}$), which can be calculated by the following formula:
$V_F = (A_U + A_D + \sqrt{(A_u + A_D)}) \times \frac{1}{3} h$               (4)
where $A_U$ and $A_D$ denote the base and top surface area of a frustum and $h$ denotes the height of the
frustum, which is 1 m in our case. In this research, RLV is calculated relative to 1989.

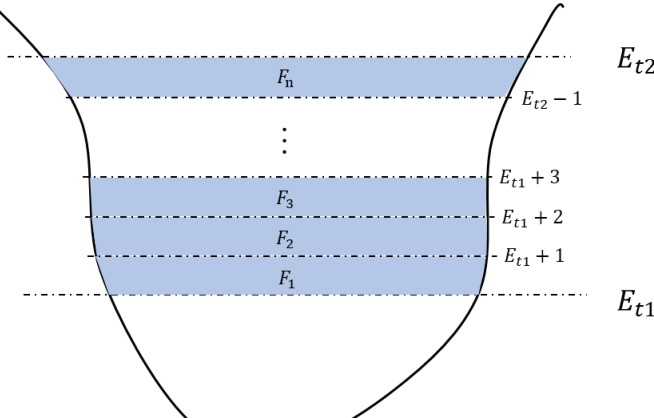

**Figure 4: Schematic diagram showing how relative lake volume can be calculated using a series of frustums.**
**The volume between time t1 to t2 can be divided into a series of frustums (F₁ to Fₙ) with a height of 1 m. For**
**each frustum, its volume can be calculated with its top and bottom area.**
**4 Accuracy assessment**
We compared our results with a widely used lake surface elevation and storage dataset from the LEGOS
Hydroweb (Crétaux et al., 2011) as well as several most recent lake volume data in the TP from Li et al.
(2019) (referred to as Li's data hereafter) and (Yao et al., 2018) (referred to as Yao's data hereafter).
Because our volume data are relative volume change to 1989, we recalculated both Li's and Yao's data
to make sure their volume data are also relative volume to 1989. Pearson's correlation coefficient (PCC)
and symmetric mean absolute percentage error (sMAPE) were used to evaluate our data, which is defined
as:
$\text{sMAPE} = \frac{1}{n} \sum_{i=1}^n \frac{2*|x_i - y_i|}{|x_i| + |y_i|}$               (5)





where n is the sample size. $x_i$ and $y_i$ are $i$th data value in our results and existing datasets, respectively.
The range of sMAPE is [0, 2] and the smaller the sMAPE, the smaller the relative error. sMAPE is a
scale-independent accuracy index based on percentage errors (Chen et al., 2017). Compared with
commonly used Root Mean Square Error (RMSE), sMAPE can be used to compare lakes with different
magnitude of RLV. In addition, sMAPE allows 0 in the data, which is very common in RLV. In contrast,
mean relative error (MRE) has issues when data values are 0. Because those reasons we used sMAPE
here.
Table 3 shows the PCC and sMAPE when comparing our results with Hydroweb (21 lakes) and Li's data
(40 lakes) for overlapping lakes. All the PCCs are significant with p-values less than 0.01. Compared
with Hydroweb data, 13 lakes (61.9%) have a PCC larger than 0.8 and a sMAPE less than 1. Compared
with Li's data, 26 lakes (65%) have a PCC larger than 0.8 and a sMAPE less than 1. Those results suggest
that our results match generally well with both Hydroweb and Li's lake data.

**Table 3: Comparison between our results and Hydroweb and Li's data. The lowest PCC and highest sMAPE**
**in each column were highlighted in italic and bold font (Lake names are from Hydroweb dataset).**

| Lake Name | Hydroweb | | Li's Data | |
|---|---|---|---|---|
| | PCC | sMAPE | PCC | sMAPE |
| Tangra-Yumco | 0.801 | 1.061 | 0.738 | 0.245 |
| Xuelian-Hu | 0.819 | 0.576 | / | / |
| Orba-Co | 0.693 | 1.195 | / | / |
| Dung-Co | / | / | 0.889 | 0.892 |
| Memar-Co | / | / | 0.954 | 0.508 |
| Pung-co | 0.970 | 0.235 | 0.983 | 0.407 |
| Yibug-Caka | / | / | 0.956 | 0.694 |
| Kyebxang-Co | / | / | 0.982 | 1.375 |
| Xuru-Co | / | / | 0.863 | 1.080 |
| Salt-Lake | / | / | 0.990 | 0.381 |
| Rola-Co | / | / | 0.995 | 0.501 |
| Salt-Water-Lake | / | / | 0.742 | 1.540 |
| Zige-Tangcuo | 0.996 | 0.422 | 0.964 | 0.693 |
| Bamco | / | / | 0.993 | 0.134 |
| Gozha-Co | / | / | *-0.118* | *1.551* |
| Donggei-Cuona-Lake | / | / | 0.945 | 0.947 |
| Zhuonai-Lake | / | / | 0.981 | 0.900 |
| Aksayqin | 0.901 | 0.684 | 0.954 | 1.129 |



| | | | | |
|---|---|---|---|---|
| Co-Ngoin1 | / | / | 0.537 | 1.048 |
| Lixiodain-Co | 0.985 | 0.730 | 0.970 | 1.011 |
| Margai-Caka | / | / | 0.966 | 0.274 |
| Dagze-Co | 0.979 | 0.323 | 0.985 | 0.267 |
| Kusai-Lake | / | / | 0.991 | 1.286 |
| Jingyu | 0.886 | 1.562 | 0.941 | 0.534 |
| Hoh-Xil-Lake | / | / | 0.975 | 0.877 |
| Lumajangdong-Co | 0.978 | 1.048 | 0.978 | 1.259 |
| Dogaicoring-Qangco | 0.954 | 0.403 | 0.901 | 0.789 |
| Urru-Co | 0.790 | 1.009 | 0.384 | 1.257 |
| Goren-Co | / | / | 0.690 | 1.326 |
| Taro-Co | 0.384 | *1.708* | 0.813 | 0.786 |
| Ngangze-Co | 0.911 | 0.299 | 0.933 | 0.199 |
| Dogia-Coring | 0.983 | 0.152 | 0.975 | 0.283 |
| Xijir-Ulan-Lake | / | / | 0.983 | 0.661 |
| Ngangla-Ringco | *-0.140* | 1.263 | 0.811 | 1.140 |
| Aqqikkol-Lake | / | / | 0.991 | 0.560 |
| Wulanwula-Lake | 0.980 | 0.307 | 0.975 | 0.329 |
| Zhari-Namco | 0.958 | 0.496 | 0.903 | 0.590 |
| Ayakkum-Lake | 0.966 | 0.968 | 0.981 | 0.981 |
| Tu-Co | / | / | 0.963 | 0.340 |
| Chibzhang-Co | / | / | 0.988 | 0.666 |
| Nam-Co | 0.935 | 0.457 | 0.918 | 0.273 |
| Selin-Co | 0.994 | 0.411 | 0.984 | 0.231 |


There are discrepancies among the datasets. For example, lake Ngangla-Ringco has a PCC of -0.140 and sMAPE of 1.263 when compared with Hydroweb data but a PCC of 0.811 and sMAPE of 1.263 when compared with Li's data. Three lakes (Ngangla-Ringco, Gozha-Co, Taro-Co), which have the largest difference from our dataset and are highlighted in Table 5, were further examined. For Ngangla-Ringco, Fig. 5 shows the differences in lake area and surface elevation between our results and the two existing datasets. From 2016 to 2019, while our and Li's lake surface elevation both show a significant increase, Hydroweb elevation has a slight decrease. And from 2002 to 2019, our lake area is around 500 km$^2$ but Hydroweb lake area is about 240 km$^2$, only about half of our lake surface area.

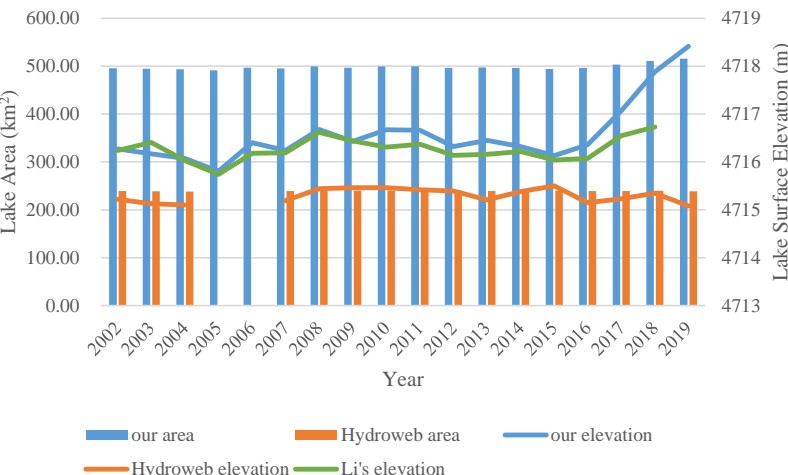


**Figure 5: Comparison of lake area and lake surface elevation between our results and two existing data**
**(Hydroweb and Li's data) for lake Ngangla-Ringco from 2002 to 2019. The y-axis on the left, representing**
**lake area, is for the vertical bars. The y-axis on the right, representing lake surface elevation, is for the lines.**





The boundaries of lake Ngangla-Ringco in 2008 (before significant increase) and 2018 (after significant
increase) are shown in Fig. 6 with SRTM DEM added to illustrate lake boundary elevation in these two
years. The mean lake boundary elevation is 4716.68 and 4717.88 meters in 2008 and 2018 respectively
and Fig. 6C-E show a distinct increase in surface elevation between the years. Our lake boundaries (Fig.
6A-B) fit well visually with the lake on the composite images, indicating our lake areas are more credible
than Hydroweb data for the lake. Although our annual composite images tend to extract the maximum
lake extent within a year, it is unlikely the lake area is twice as large as that in Hydroweb.

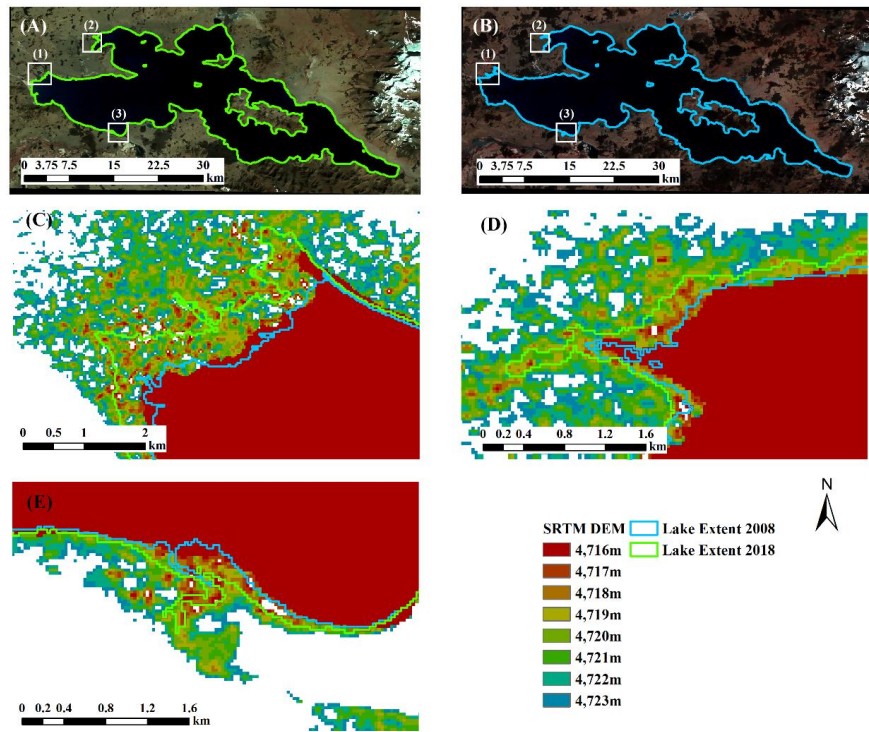

**Figure 6: Lake extents in 2018 (A) and 2008 (B) and in three close-up areas (C), (D) and (E) (corresponding to boxes (1), (2), (3) in (A) and (B), respectively) from our results for lake Ngangla-Ringco. Images in (A) and (B) are composite image (R: Near-infrared band, G: Red band, B: Green band ) from Landsat 5 and Landsat 8 respectively. DEM shown in (C)-(E) are SRTM DEM.**

Lake Gozha-Co showed distinct trends in lake surface elevation and volume between our results and Li's data (Fig. 7). In Li's data, lake surface elevation rose from 2001 to 2009 with the highest elevation of 5084.43 m in 2009, and then started a decrease trend. In our results, lake surface elevation fluctuated but generally had been decreasing from 2001 to 2018. While our results have an elevation range between 5079 and 5081 m, the elevation range of Li's data is between 5083 and 5085 m, which leads to extremely larger lake volume compared with our data.



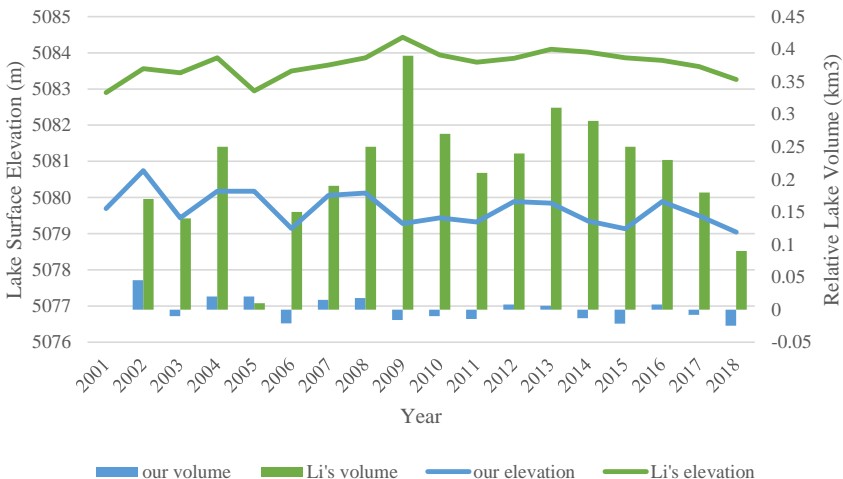

321

**Figure 7: Comparison of relative lake volume and lake surface elevation between our results and two existing data (Hydroweb and Li's data) for lake Gozha-Co from 2001 to 2018. The y-axis on the left, representing ake surface elevation, is for the lines. The y-axis on the right, representing raletive lake volume, is for the vertical bars.**

326

For further assessment, extracted extents (Fig. 8A-C) for lake Gozha-Co in 2002, 2009, and 2018 and

SRTM DEM are shown in Fig. 8. The mean lake boundary elevation is 5080.74, 5079.28 and 5079.04

meters in 2002, 2009 and 2018 respectively and Fig. 8D-E show no distinct change in surface elevation,

confirming our surface elevation is more reliable. Fig. 8 also shows that the highest lake surface elevation

occurred in 2002 rather than 2009, and the lake surface elevation in 2009 and 2018 did not differ much.

The large difference in volume might be caused by the gaps in elevation. But a definite conclusion cannot

be drawn as Li's data doesn't provide lake area information.

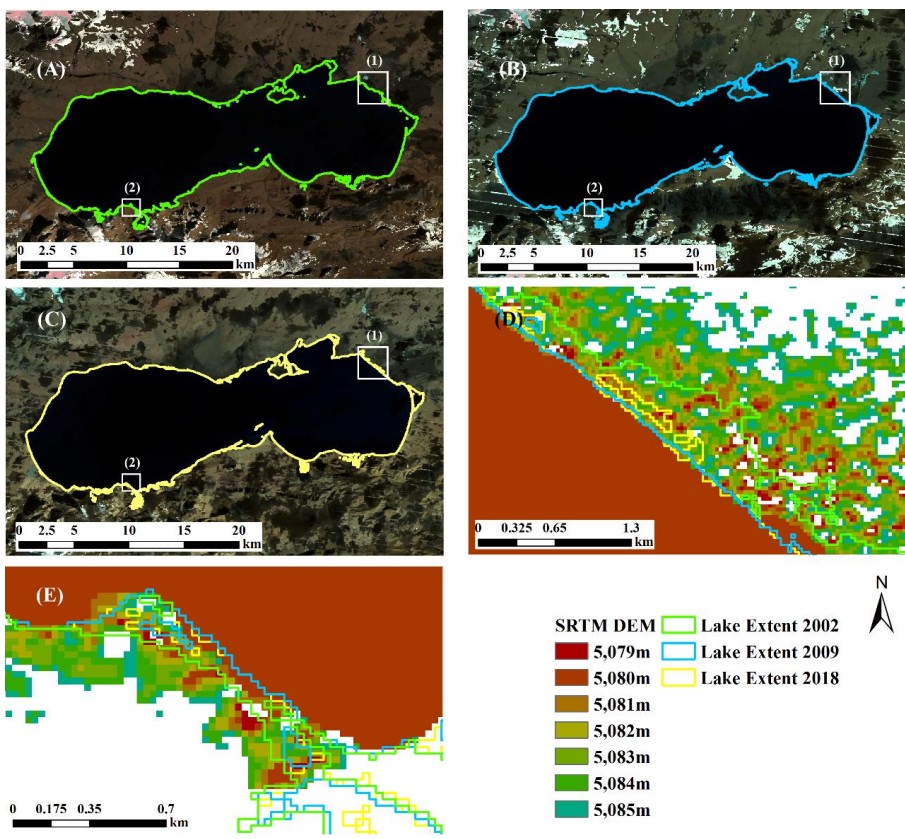

**Figure 8: Lake extents in 2002 (A), 2009 (B), and 2018 (C) and two close-up areas (D), and (E) (corresponding to boxes (1) and (2) in image (A), (B) and (C), respectively) from our results for lake Gozha-Co. DEM shown in (D) and (E) are SRTM DEM . Composite images in (A)-(C) (R: Near-infrared band, G: Red band, B:Green band) are from Landsat 7.**

Figure 9 shows the differences in lake area and surface elevation among the datasets for lake Taro-Co. Our results and the two existing datasets generally have a similar increase trend in surface elevation in 2004-2008. In our results, surface elevation had been increasing from 2015 to 2018 but Hydroweb elevation experienced a decrease from 2015 to 2016 and Li's elevation had also been decreasing from 2017 to 2018. In addition, both our area and elevation fluctuated more than the other two datasets.

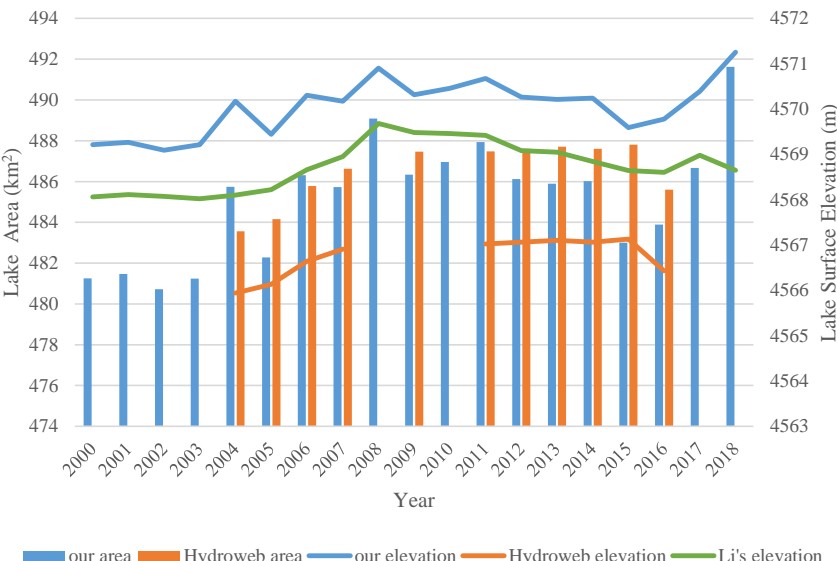

346

**Figure 9: Comaprison of lake area and lake surface elevation between our results and two existing data**
**(Hydroweb and Li's data) for lake Taro-Co from 2000 to 2018. The y-axis on the left, representing lake area,**
**is for the vertical bars. The y-axis on the right, representing lake surface elevation, is for the lines.**

350

For further assessment, the extracted extents (Fig. 10A-C) for lake Taro-Co in 2015, 2016, and 2018 and

SRTM DEM were shown in Fig. 10. Our lake boundaries visually fit well with lake extents on the

composite images and the mean elevation of the lake boundaries is 4569.59 m, 4569.77 m, and 4571.25

m, respectively. A significant increase in lake surface elevation in 2018 can be clearly observed in Fig.

10D-E.

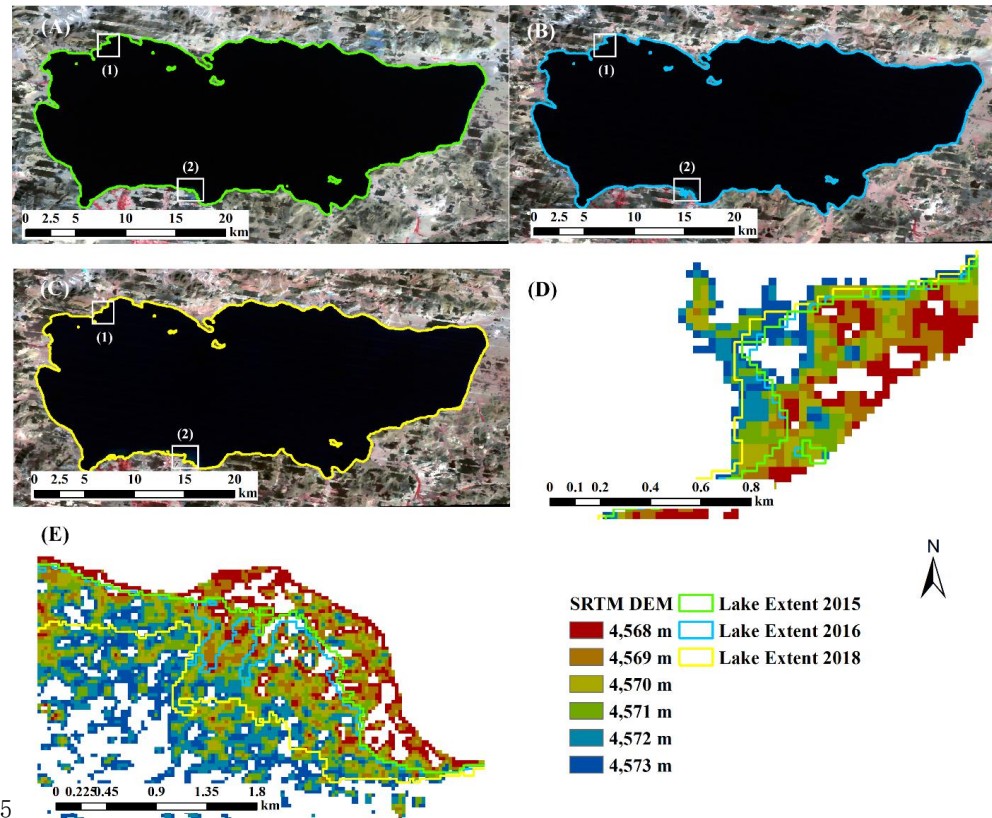

**Figure 10: Lake extents from our analysis for lake Taro-Co in 2015 (A), 2016 (B), and 2018 (C) and two close-up areas (D), and (E) (corresponding to boxes (1) and (2) in image (A), (B) and (C), respectively). DEM shown in (D) and (E) are SRTM DEM . Composite images in (A)-(C) (R: Near-infrared band, G: Red band, B:Green band) are from Landsat 7.**

Yao et al. (2018) also published a lake storage data in the IB. Their datasets include the annual RLV for

871 lakes with an area larger than 1 km$^2$ from 2009 to 2015, and the annual RLV for 126 lakes with an

area larger than 50 km$^2$ from 2002 to 2015. We found 816 overlapping lakes from 2009 to 2015 and all

the large lakes (126) in our dataset. The main reason that our dataset has less lakes in the IB is that

connected waterbodies were counted as separate lakes in Yao's data (as shown in Fig. 11).



367

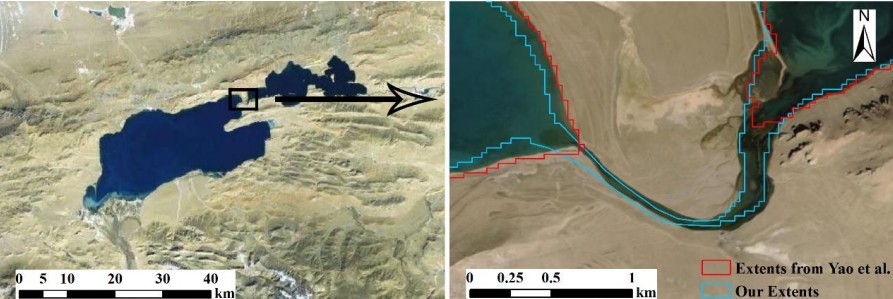

**Figure 11: An example that connected waterbodies were counted as separate lakes in Yao's data. Remote sensing image is from http://t0.tianditu.gov.cn/img_c/wmts. Background remote sensing image is from http://t0.tianditu.gov.cn/img_c/wmts.**

The PCC and sMAPE for the overlapping lakes (816) are shown in Table 4. For lakes larger than 1 km$^2$, when the p-value is greater than 0.05, the PCCs of all lakes are less than 0.8, and 84.01% of lakes have sMAPE greater than 1. This means that for these 371 lakes, there is a big difference between our results and Yao's data. There are 389 lakes (47.67%) have a PCC greater than 0.8 and a p-value less than 0.05, and 71.91% of lakes have a sMAPE less than 1. This means that for these 445 lakes, our results have high consistency with Yao's. For lakes with an area greater than 50 km$^2$, 109 out of 126 (86.51%) lakes have p-value less than 0.05. For lakes with p-value less than 0.05, 86 out of 109 (78.90%) lakes have PCC larger than 0.8 and 73.40% lakes have sMAPE less than 1. Overall, most of our lake data match well with Yao's data. Because Yao et al. (2018) did not provide lake area and surface elevation data, it is difficult for us to further examine the discrepancy.

**Table 4: Data comparison statistics between our results and Yao's data.**

| Dataset | p-value | Total | PCC <0.6 | 0.6≥PCC<0.8 | PCC≥ 0.8 | sMAPE <1 | sMAPE≥1 |
|---|---|---|---|---|---|---|---|
| **Lake area > 1 km$^2$** | > 0.05 | 371 | 251 | 120 | 0 | 84 | 287 |
| | ≤ 0.05 | 445 | 5 | 51 | 389 | 320 | 125 |
| **Lake area > 50 km$^2$** | > 0.05 | 17 | 17 | 0 | 0 | 2 | 15 |
| | ≤ 0.05 | 109 | 3 | 20 | 86 | 80 | 29 |




In summary, our results generally show a high consistency with the existing datasets, though large
discrepancy does exist for some of the lakes. Close examination on a few extreme lakes indicated that
our results are more reliable and more in line with Landsat imagery and SRTM DEM.
**5 Results**
We identified a total of 976 lakes in the EBTP, and their maximum extents during the study period are
shown in Fig. 12. 930 of those lakes (95.29%) are located in the Inner Basin, and only 46 (4.71%) are in
the Qaidam Basin. Large lakes are primarily located in the southern and eastern periphery of the inner
basin.
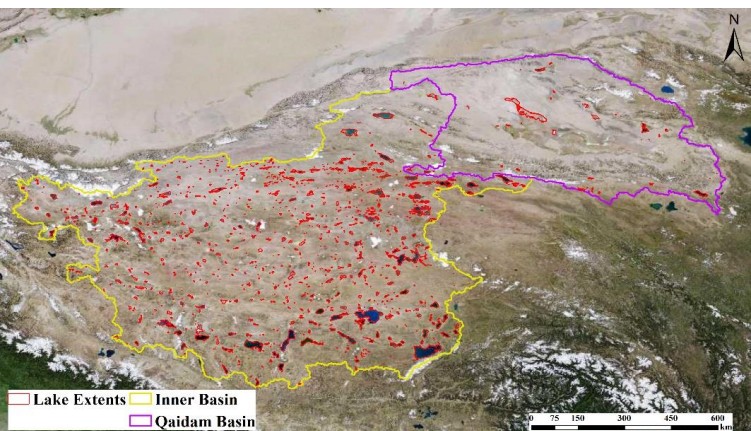

**Figure 12: A total of 976 lakes larger than 1 km² identified in the EBTP. Remote sensing image is from**
**http://t0.tianditu.gov.cn/img_c/wmts.**
**5.1 Lake water volume change**
Total lake volume in the study area exhibited a net increase of 193.45 km³ from 1989 to 2019 with an
increase rate of 6.45 km³ year⁻¹. Although lake volume was generally increasing in the past 30 years, it
varied significantly from year to year. Figure 13 shows annual total loss, gain, and net change of lake
volume from 1989 to 2019. The lakes experienced water gain in 23 years and loss only in 7 years in the
30 years of study period. From 1998 to 2013, the lakes experienced the longest continuous water gain of
16 years. The largest water gain of 25.19 km³ appeared in 2000, and the largest water loss of -18.15 km³
occurred in 1994.

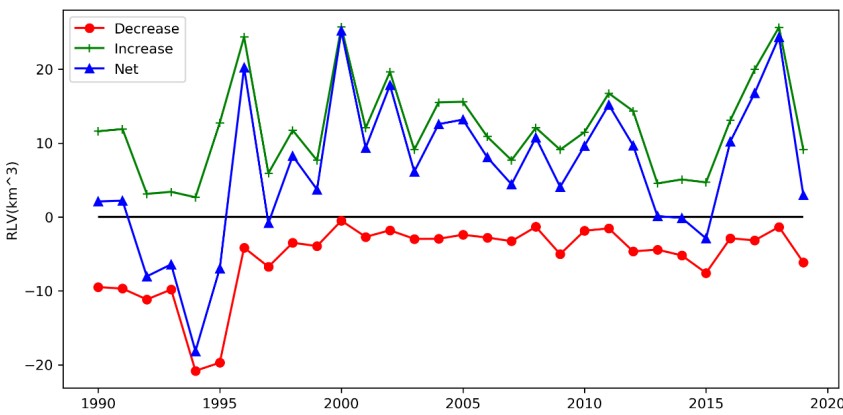

**Figure 13: The gain, loss and net lake water volume in the EBTP from 1989 to 2019.**

Figure 14 shows the trend of annual RLV in the entire study period and in 7-year periods (1989-1995,
1995-2001, 2001-2007, 2007-2013, 2013-2019) at each lake. Positive trend slope represents an overall
increase in lake volume and vice versa. Similar to some previous studies (Yao et al., 2018; Zhang et al.,
2017b), 909 lakes (93.14%) had been expanding in the study period with the exception of 67 lakes
(6.86%). 16 lakes gained more than 0.1 km$^3$ of water per year, and these lakes are mainly located in the
east side of the IB (Fig. 14A).
RLV trend varied in the 7-year time periods. From 1989 to 1995, only 418 lakes (42.83%) experienced
volume expansion, and in fact, a noticeable lake shrinkage is observed from 1989 to 1995 (Fig. 14B)
where most lakes have a decreasing trend and lakes with large RLV decrease (> 0.1 km$^3$ per year) are
mostly located on the east or west side of IB. From 1995 to 2001 (Fig. 14C), 816 lakes (83.61%) had
been expending. While most lakes in the QB were still decreasing, most lakes in the IB had increase
trend with large RLV increase (>0.1 km$^3$ per year) mostly located at the north, east and south periphery
of the IB. From 2001 to 2007 (Fig. 14D), though the changing trend is similar to 1995-2001, the increase
rate got smaller as there are more yellow lakes than light green lakes in Fig. 14D, indicating more lakes
have negative changing rate (-0.05-0km$^3$/y) in 2001-2007. The increasing trend in 2007-2013 (Fig. 14E)
is very similar to the previous period but with a lower rate as there are less large increase lakes (dark blue
lakes) and a couple of large decrease lakes (red lakes). From 2013 to 2019 (Fig. 14F), strong increasing
trend occurred again with more blue lakes in both IB and QB.



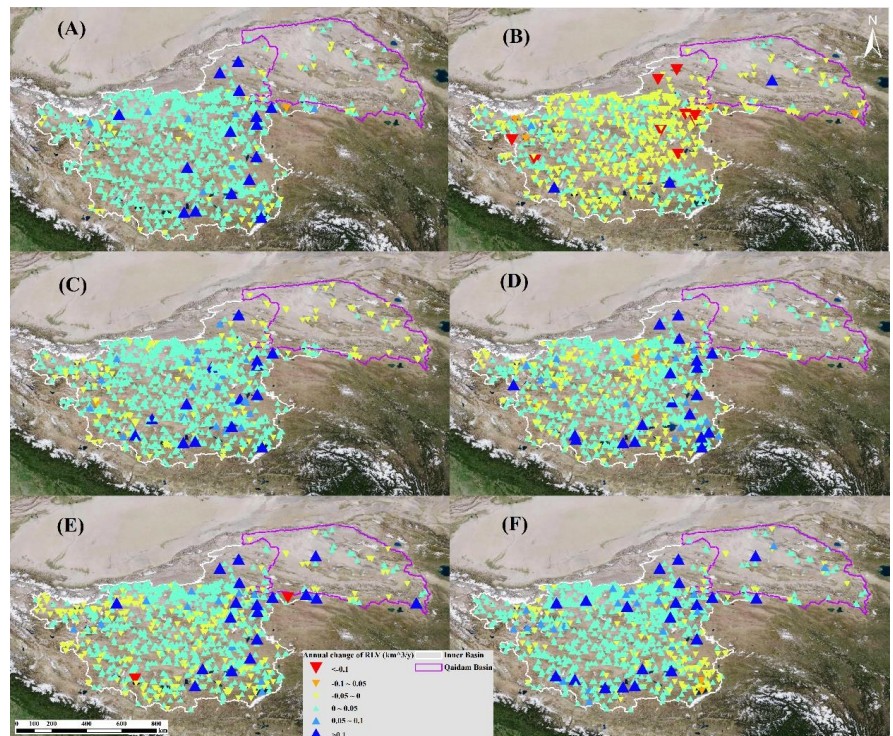

**Figure 14: Trend of annual RLV during the periods of (A) 1989-2019, (B) 1989-1995, (C) 1995-2001, (D) 2001-2007, (E) 2007-2013, and (F) 2013-2019. Background remote sensing image is from http://t0.tianditu.gov.cn/img_c/wmts.**

Trend analysis was performed for the EBTP, its sub-regions (IB, QB) and different time periods. The slope and coefficient of determination ($R^2$) are shown in Table 5. It suggests that there was a significant increasing trend both in the TP and IB in the recent 30 years. While the trend slope is positive in the QB (0.0700), it is much smaller than that of EBTP (7.28) and IB (7.45). $R^2$ in the QB is 0.242 and it's significant at 0.01 confidence level indicating a weak increasing trend. Trends in the IB and EBTB are similar in the 7-year periods but this is not the case in the QB. This is mainly due to that most of the lakes are located in the IB. Trend slopes in Table 5 correspond well to Fig. 14 which indicate that the entire EBTB experienced a lake volume decrease (slope=-6.47, $R^2$=0.800) in 1989-1995. In 1995-2001, IB's lake volume increased (slope=10.23, $R^2$=0.925) while QB's lake volume decreased (slope=-0.153, $R^2$=0.708). From 2001 to 2019, although the overall volume of lake water has been increasing, the slope in 2007-2013 (8.93) was less than that in 2001-2007 (10.43) and 2013-2019 (9.92).





443

**Table 5: Trend of total RLV in EBTP and its sub-region IB and QB in different time periods (\* indicates significant at a confidence level of 0.01).**

| Time Period | Index | EBTP | IB | QB |
|---|---|---|---|---|
| 1989-2019 | Slope (km$^3 \cdot$year$^{-1}$) | 7.28 | 7.45 | 0.0700 |
| | R² | 0.921* | 0.923* | 0.242 |
| 1989-1995 | Slope (km$^3 \cdot$year$^{-1}$) | -6.47 | -6.29 | -0.174 |
| | R² | 0.800 | 0.797 | 0.631 |
| 1995-2001 | Slope (km$^3 \cdot$year$^{-1}$) | 10.08 | 10.23 | -0.153 |
| | R² | 0.921 | 0.925 | 0.708* |
| 2001-2007 | Slope (km$^3 \cdot$year$^{-1}$) | 10.43 | 10.28 | 0.156 |
| | R² | 0.978* | 0.979* | 0.439* |
| 2007-2013 | Slope (km$^3 \cdot$year$^{-1}$) | 8.93 | 8.59 | 0.343 |
| | R² | 0.969* | 0.966* | 0.420 |
| 2013-2019 | Slope (km$^3 \cdot$year$^{-1}$) | 9.92 | 9.49 | 0.422 |
| | R² | 0.842* | 0.850* | 0.300 |

446

**5.2 RLV and lake area**

Figure 15 shows annual RLV trend slope by lake area. For most lakes in 1 - 10 km², their RLV trend slope is between 0 and 0.003 km³/y, indicating slow increase in water volume in the past 30 years. As lake area increases from 10-50 km² to greater than 50 km², RLV trend slopes also increased (Fig. 15C-D) though the number of lakes reduced. Nevertheless, there are some exceptions. For example, there are lakes with area larger than 100 km² (Fig. 15D) but their RLV increasing rate is less than 0.003 km³/y. Some lakes with an area between 10-50 km² have annual RLV larger than 0.01 km³/y (Fig. 15B). Some small lakes, with an area less than 10 km², have decreasing RLV rate smaller than 0.01 km³/y (Fig. 15A).

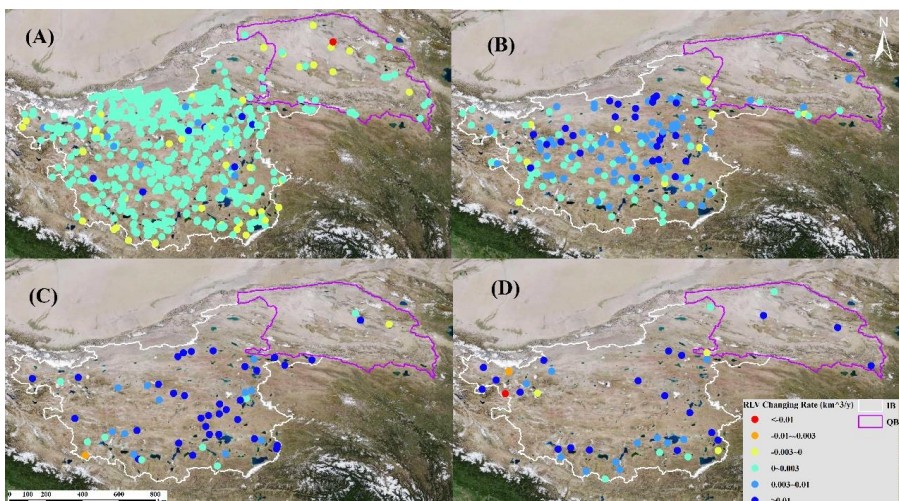

**Figure 15: Annual RLV trend by lake area of (A) 1 - 10 km², (B) 10 - 50 km², (C) 50 - 100 km², (D) > 100 km².**

**Background remote sensing image is from http://t0.tianditu.gov.cn/img_c/wmts.**

Table 6 shows the statistics of annual RLV trend by lake area. In general, the larger the lake area, the

greater the trend slope. The mean and standard deviation of the trend slopes both increase with the

increase of area. The range of the RLV rate for lakes of 0-10 km² is larger than that for lakes of 10-50

km², indicating extreme changes occurred in smaller lakes

**Table 6: Statistics of annual RLV changing rate.**

| Statistics of | Lake Area | | | |
|---|---|---|---|---|
| Annual RLV changing rate | 1 - 10 km² | 10 - 50 km² | 50 - 100 km² | > 100 km² |
| Count | 675 | 175 | 56 | 70 |
| Minimum (km³/y) | -0.038 | -0.0014 | -0.0054 | -0.051 |
| Maximum (km³/y) | 0.037 | 0.037 | 0.085 | 1.04 |
| Mean (km³/y) | 0.00068 | 0.0052 | 0.016 | 0.075 |
| Standard Deviation (km³/y) | 0.0032 | 0.0059 | 0.015 | 0.15 |

**6 Discussions**

**6.1 Methods for deriving lake elevation-area relationship**

Lake surface elevation can be estimated by calculating the average elevation of lake boundary (Bao et

al., 2005; Li et al., 2019; Yang et al., 2017a; Yao et al., 2018). This approach assumes that the DTM are

obtained before lake water volume starts increase. The DTM we used were acquired in and after 2000



(Takaku et al., 2014; Van Zyl, 2001) but our study period starts from 1989. As such, lake surface
elevation in this study is estimated based on the area-elevation relationship derived from the DTM.
Existing studies mainly used just one of a few methods, including linear equation (Yang et al., 2017b),
parabolic equation (Li et al., 2019) or monotonic cubic spline fitting (Yao et al., 2018), in deriving lake
elevation-area relationship. In this research, we compared those methods and used different methods
under different situations (see Sect. 3.4).
Four lakes, with area ranging from 0.97 km$^2$ to 149.3 km$^2$, were selected to explain the typical situations
when different methods were used. Figure 16 shows the elevation-area pairs (red points) from the DTM
and estimated elevations based on image lake area using different data fitting methods. For fitting the
data from the DEM, MCI has the best fitting performance for the lakes in Fig. 16A & B and there is no
obvious disparities between SOPR and MCI in Fig. 16 C & D. The LR has the worst performance in Fig.
16 A, B & C. However, when the elevation-area pairs from the DTM do not cover the lake area range
from Landsat images, estimated elevation can have serious error, especially for MCI. Take the lake in
Fig. 16B as an example, its area range from Landsat imagery is [0.23, 16.71] km$^2$ from 1989 to 2019, yet
the smallest area obtained from SRTM is 4.69 km$^2$. This is because the DTM were obtained after 2000
but most lakes had been expending since 1995 in the region. While MCI fits well with the elevation-area
data from the DTM, elevations estimated outside the DTM area range are unreal in Fig. 16B & D),
especially in Fig. 16D, where the elevation estimates for lake area smaller than the smallest area from
the DTM are unreasonably high. Those examples indicate that MCI may overfit and should only be used
for lakes when their image area is within the area range from the DTM. SOPR predicted lake elevations
generally follow the same trend when lake image area is outside DTM area range. As such, SOPR is
selected when lake image area is smaller than the minimum area from the DTM. In addition to the above
situations, the number of elevation-area pairs from the DTM within the area range of [MinA/1.5,
MaxA*1.5] also play a role as discussed in Sect. 3.4. Besides, some other situations also affect the choice
of the methods. When using SOPR method, the fitted curve is not monotonic if its symmetric axis falls
into [MinA, MaxA] (Fig. 16A). When this happens, LR method was used instead.

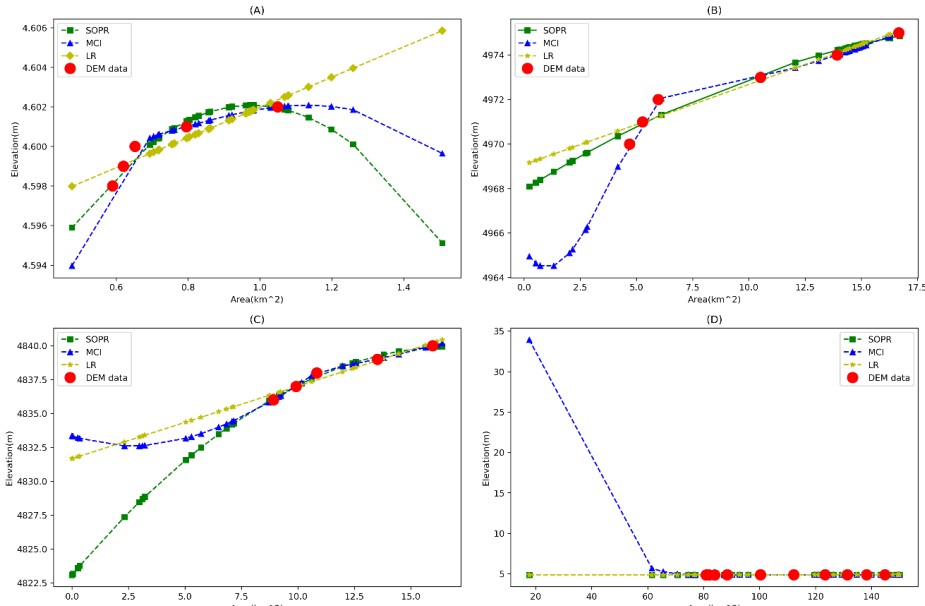

**Figure 16: Estimated elevation based on image lake area using LR, SOPR and MCI. The elevation-area data pairs obtained from SRTM DEM is also added.**

The number of lakes and the minimum, maximum, and average lake area for each method are listed in Table 7. The most used method is SOPR with 766 lakes. While LRN and LRC are typically used for small lakes, MCI is selected mostly for large lakes. Since MCI was only used for lakes when their image area is within the area range from the DTM, this indicates most large lakes' area started increasing after 2000. In summary, we found no single method is suitable for all the lakes, and different methods have to be used for different lakes.

**Table 7: Frequency and lake area statistics for each method used in deriving the lake elevation-area relationship. Lake area is for 2019.**

| Methods | Frequency | Minimum lake area (km²) | Maximum lake area (km²) | Average lake area (km²) |
|---------|-----------|--------------------------|--------------------------|--------------------------|
| LRN | 24 | 0.049 | 27.35 | 3.40 |
| LRC | 30 | 0.86 | 9.72 | 2.30 |
| LRS | 75 | 0.028 | 1044.80 | 62.74 |
| SOPR | 766 | 0.049 | 1078.81 | 25.71 |
| MCI | 81 | 1.46 | 2016.52 | 121.83 |

**6.2 RLV variation**

Although lakes with larger area usually have larger RLV trend slope, we found that the range of the

change rates for the lakes in 1 - 10 km$^2$ is larger than that for the lakes in 10-50 km$^2$ in Sect. 4.2. Here

we further examined the relationship between lake area and the coefficient of variation (CV) of RLV

(Fig. 17). While there is lack of correlation between them, the percentages of lakes with |CV| >10 in the

four area ranges are 3.7%, 2.3%, 1.8%, and 2.9% respectively, with lakes in 1 – 10 km$^2$ having the highest

ratio. The lakes with extreme RLV are mostly located in the peripheral of the IB and QB (Fig. 17).

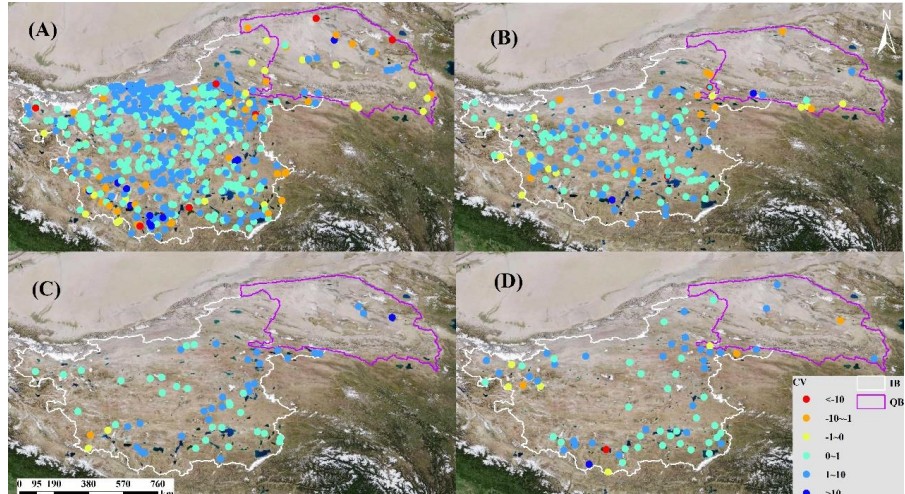

**Figure 17: CV of annual RLV by lake area of (A) 1 - 10 km$^2$, (B) 10 - 50 km$^2$, (C) 50 - 100 km$^2$, (D) > 100 km$^2$. Background remote sensing image is from http://t0.tianditu.gov.cn/img_c/wmts.**

The minimum, maximum and mean CV of all lakes are -106.65, 82.77 and 0.89, respectively. And 94.36%

(921 out of 976) of the lakes have a CV between -1~1, which indicates that the remaining few lakes have

significant volume changes in the past 30 years. The annual lake area and RLV of the three lakes with

the highest absolute CV are shown in Fig. 18. All three lakes have significant volume fluctuation in the

past 30 years. For lake (1) (Fig. 18C(1)), its volume decreased significantly from 1994 to 1996 and

increased rapidly from 2003 to 2009. For lake (2) (Fig. 19C(2)), its volume fluctuated cyclically in the

past 30 years. From 1989 to 1996, its water volume had been continuously decreasing and reached the

minimum RLV of -0.0011 km$^3$. From 1996 to 2004, its lake volume kept rising and reached the maximum

RLV of 0.0026 km$^3$. Subsequently, its volume started to decline again, reaching a minimum value of -

0.0013 km$^3$ in 2017. For lake (3)(Fig. 19C(3)), its volume had been expanding slowly after 2000.



However, between 1990 and 2000, its volume fluctuated significantly. While all these example lakes are
in 1 - 10 km² and have extreme CVs, their temporal variations are different indicating the influence of
local hydro-climatic factors on lake dynamics

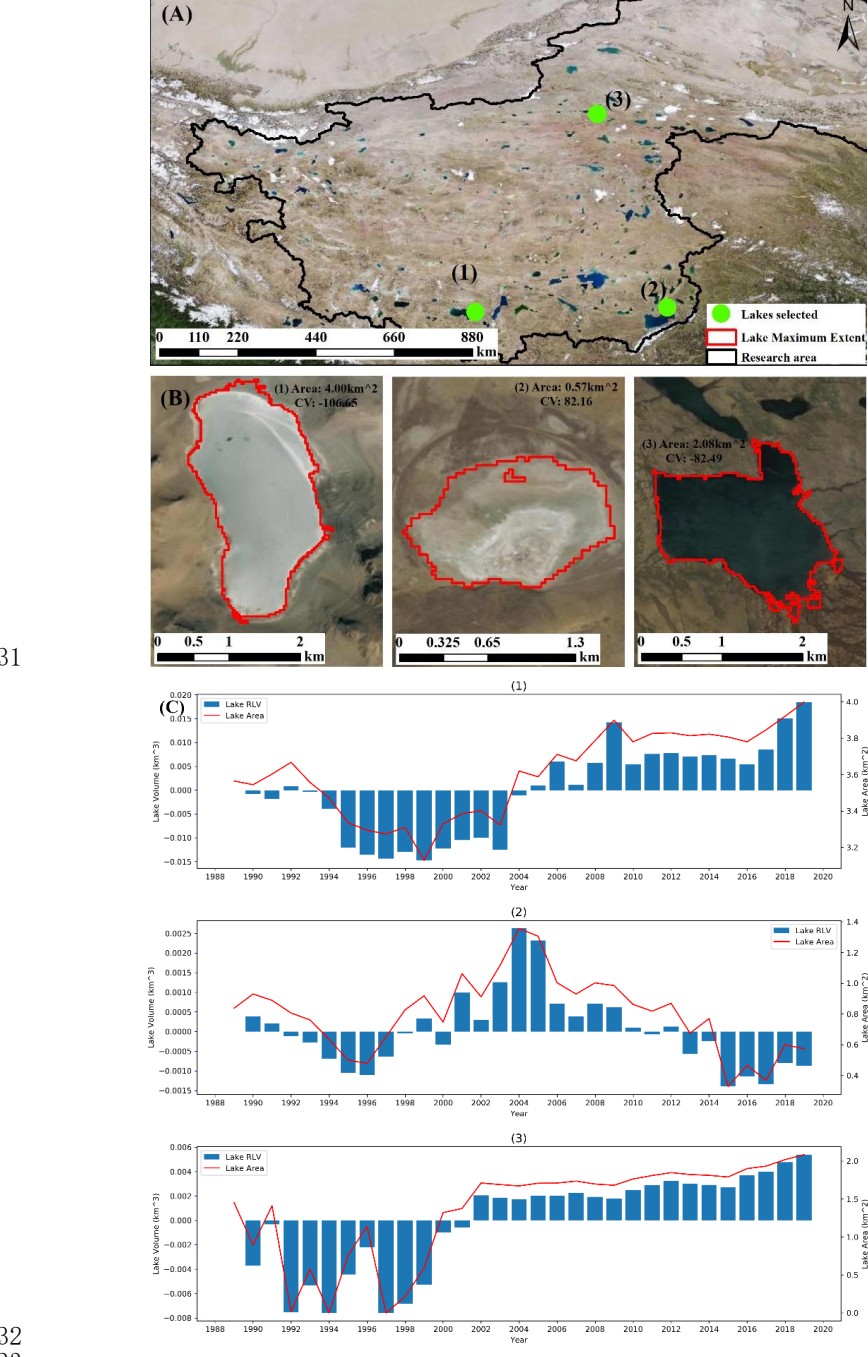



**Figure 18: The location of three lakes with the highest absolute CV in the EBTP (A), their maximum extents**
**(B) and area and RLV time series of the lakes (C). Remote sensing images in (A) and (B) are from**
**http://t0.tianditu.gov.cn/img_c/wmts.**





In previous research, although some studies (Li et al., 2019; Zhang et al., 2020; Dong et al., 2018) have
found that lakes of different sizes respond differently to climate change, there is a lack of attention to
lakes less than 10 km$^2$. This is mainly due to insufficient data of small lakes in the region. Our results
indicate that small lakes are important as lakes with higher CV are usually less than 10 km$^2$, which,
however, is different from the definition of small lakes in Zhang et al. (2020) (50-100 km$^2$) and Dong et
al. (2018) (10-30 km$^2$). Our results show that lakes less than 10 km$^2$ are more prone to drastic volume
change and should receive more attention. In addition, most existing data products focused on lake area
instead of volume change, though RLV is more valuable in studying water balance in hydrological
systems.
**7 Conclusions**
This research provides a comprehensive census on water volume change for the lakes greater than or
equal to 1 km$^2$ in the EBTP from 1989-2019 using Landsat imagery and DTM data. Our annual dataset,
compared with satellite altimetry and other existing data, covers more lakes, especially small lakes in 1
– 10 km$^2$, and longer time period.
The comparison with three other major existing data products indicates that our dataset is reliable and
might be more accurate. To the best of our knowledge, our dataset provides the longest and most
comprehensive lake water volume change data in the region, especially for small lakes (1-10 km$^2$). The
dataset is valuable in studying the impacts of climate change and water balance in the region.
Our research indicates that small lakes with an area in 1 - 10 km$^2$ are most sensitive and have the highest
fluctuation in water volume in the study time period. Monitoring their changes is of critical importance
for understanding regional and global climate change. In deriving the lake area-elevation relationship
from DTM, the best result comes from the combination of several data fitting methods. The workflow
used in this research can be further developed to process individual remote sensing image (instead of
annual composite image) and create a lake volume dataset with a higher temporal resolution in future
research.



**8 Data availability**

We completed a census of annual lake area and volume change for 976 lakes larger than 1 km$^2$ in the

endorheic basin of the Tibetan Plateau (EBTP) during 1989-2019 using Landsat imagery and digital

terrain models. This dataset consists of two lake extents shapefiles containing the annual area and

relative volume data from 1989 to 2019 for each lake. In addition, the lake seeds used to identify the

lakes are also included as a shapefile in this dataset. The dataset

(https://doi.org/10.5281/zenodo.5543615, Wang et al., 2021), entitled "Lake area and volume variation

data in the endorheic basin of the Tibetan Plateau from 1989 to 2019", is available on Zenodo.

**Author contribution**

Conceptualization, L.Z. and X.L.; methodology, L.W., X.L.; software, M.L.; validation, L.W., M.L. and

J.W.; formal analysis, M.L.; investigation, L.W.; resources, J.W.; writing—original draft preparation,

L.W.; writing—review and editing, J.W.; visualization, M.L.; supervision, X.L.; project administration,

J.W.; funding acquisition, L.Z. All authors have read and agreed to the published version of the

manuscript.

**Competing interests**

The authors declare that they have no conflict of interest.

**Acknowledgements**

This work was supported by the Chinese Academy of Sciences Strategic Priority Research Program No.

XDA20020100, by the National Natural Science Foundation of China under Grant No. 41771243, and

by the Open Fund Project of the Key Laboratory of Coastal Zone Exploitation and Protection, Ministry

of Natural Resources No. 2019CZEPK01.

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
