# Peer review of "Lake area and volume variation in the endorheic basin of the Tibetan Plateau from 1989 to 2019"

_Earth System Science Data, 2021_

## Referee Comment (RC2)

The readability of this manuscript is very poor. The terminologies and conceptions are not clearly explained. The data description should be more detail, and the descriptions of the method are not logically organized. Besides, many sentences have no logicality and should be rewritten. Authors are strongly suggested to reorganize this paper and ask for an English edition.

Abstract:
L13: it is not a region susceptible to climate change, but something in this region susceptible to climate change. please rewrite this sentence to make sure right logic.
L20: lake volume means the water volume or the size of lake?

L21-22: the meaning of "the larger the lake area, the greater the lake voloume change" is opposite to the meaning of " lakes with an area less than 10km2 have more severe volume change". please rewrite these two sentence to make sure consistence expression, based on the study results.

L27: what is relative lake volume?
Please add lake area in keywords.

Introduction:
L63L71 comprehensive census is not suitable for this study. Here, only lake area and volume. Please use more suitable word.

L85: was-were
L89: what are their analysis extents?
L90: delineate lake's approximate extents from JRC-GSW: do you mean correcting the extents from JRC-GSW?

3methods
It is difficult to clearly obtain information from figure 1. It is suggested to add an overall flowchart instead of figure 1.

L101: estimate lake surface elevation from lake area: delete from lake area
L110: analysis extent and seed determination are not suitable in the title of section 3.1
L114: inside a lake's analysis extents: analysis was not conducted for throughout the whole area of a lake?

L115-117: what is water cover frequency? What is water occurrence band? Because of no background introduction, the meaning of this sentence is confused.

L118: Delete "Some of those"
L120: "the border between lakes and rivers is hard to define". Is it used to classify lakes from rivers, or the separate neighboring lakes and rivers?

Do you derive the lakes from JRC-GSW every year? Does the number of lakes change from 1984 to 2019?

the total 976 lakes : 976 is the average number of several decades?

Fig.3: it is better to presents rivers and lakes in a picture using different color line.

In this study, approximate extent was frequently used. Authors want to say the extent is not precise. No one can obtain the just right number of the extent. The "approximate " is suggested to delete.

L140: Just keep one of "identify" and "distinguish"

L141-143: what is lake seed? What does it used for? How to determine the centroid point?

L148-149: "around each extent" means "around each lake"?
Could you please added a picture to explain the analysis extent?

L154-162: The meaning of this paragraph is unclear. Please reorganize it. Please note the use of terminology, for example: annual max-water
How to select images to get composite image? What is composite image? Is it mean mosaic image?
Do you get multiple composite images in a year or only one composite image?

The JRC-GSW lakes were also derived from Landsat data. What is the difference between the method adopted by JRC-GSW and in this study?

L171: "more robust in······"-"more robust than the other two method in······"

There are many steps and method in the segment process, so I suggest add a flowchart to make the process clear.

L184: lake's analysis extent: before section 3.3, there are two buffer extent. Which one do you use in this section?
L186: what is lake's seed?

From the presentation of L183-189, it is difficult to understand how to get annual lake area.

L195: significant change ? could you give a threshold?
L196: How to determine if the image is contaminated or unreliable?

L200-204: since satellite altimetry was not used here, it is not necessary to introduce it.
Directly describe DTM, and present its coverage and spatial and temporal resolution.

L205-219: It is suggested to simplify the description of the three methods. But, describe how to combine the tree method in detail.

L219: based on- over
L221: "At each elevation······ are identified". The meaning of the sentence is unclear

So far, it is unclear what is maximum lake water extent. The yearly maximum for each lake or the maximum lake of all lakes?

L226: the list with more elevation-area pairs: the comparison was conducted between SRTM and ALOS?

The lake surface elevation derivation process presented in Section 3.4 is too confused to understand. The readability is poor.

L243-244: delete "while it is impossible······bathymetry data"
Rewrite "we can calculate······dates."

Section 3.5 is hard to understand. Could you please present the definition of RLV first?

L203, 258, 265, 268 etc. please avoid using our study, our results, our volume data. You can use this study, the data in this study etc instead. Check it throughout the manuscript and make revision.

Section 4:
L268: "relative volume change to 1989". So is lake volume in section 3.5 the difference of water volume between at current time and in 1989 ?

L273: x and y present this results and results in existing studies. What are the results? Volume change or volume?

L283-284: "Those results······Li's lake data." Please check the logic of this sentence. The results match well with data.
L291: "three lakes, ······table 5". Lakes have difference from dataset
There are many sentence with wrong logic throughout this manuscript. You are strongly suggested to ask for an English edition.

L304: what is difference between lake boundary elevation and lake surface elevation?

Section 4 accuracy assessment
It is suggested to divide this section into 2 sub-section. The assessment on lake area, and the assessment on volume.
Do extent and area have the same meaning in this study? If do, please unify it throughout the manuscript.

Still the question, during 1989 to 2019, did the number of lakes not change?

In section 5.1, the results for RLV trends are introduced. In Section 5.2 it is also described. In text, RLV and lake water volume change are used. Are they different?

Why are there no results for the lake area change in section 5? The title is "lake area and volume variation"
There are many abbreviations. It is suggested to list them in appendix.

Section 6.2 points out the variation of several lakes. The content is suggested to move to results.

L546: "This research provides a comprehensive census on water volume change for the lakes". This study only provide water volume change? this expression does not fit with the title.

L552: please note the usage of comprehensive